# Regulation of BDNF-TrkB Signaling and Potential Therapeutic Strategies for Parkinson’s Disease

**DOI:** 10.3390/jcm9010257

**Published:** 2020-01-17

**Authors:** Wook Jin

**Affiliations:** Laboratory of Molecular Disease and Cell Regulation, Department of Biochemistry, School of Medicine, Gachon University, Incheon 21999, Korea; jinwo@gachon.ac.kr

**Keywords:** TrkB, Parkinson’s disease, BDNF, TrkB isoform, neuronal degeneration

## Abstract

Brain-derived neurotrophic factor (BDNF) and its receptor tropomyosin-related kinase receptor type B (TrkB) are widely distributed in multiple regions of the human brain. Specifically, BDNF/TrkB is highly expressed and activated in the dopaminergic neurons of the substantia nigra and plays a critical role in neurophysiological processes, including neuro-protection and maturation and maintenance of neurons. The activation as well as dysfunction of the BDNF-TrkB pathway are associated with neurodegenerative diseases. The expression of BDNF/TrkB in the substantia nigra is significantly reduced in Parkinson’s Disease (PD) patients. This review summarizes recent progress in the understanding of the cellular and molecular roles of BNDF/TrkB signaling and its isoform, TrkB.T1, in Parkinson’s disease. We have also discussed the effects of current therapies on BDNF/TrkB signaling in Parkinson’s disease patients and the mechanisms underlying the mutation-mediated acquisition of resistance to therapies for Parkinson’s disease.

## 1. Introduction

Neurotrophins (NTs) are growth factors, which are critical mediators for the survival and development of neurons of the peripheral and central nervous systems (CNS), through their tropomyosin-related kinase (Trk) receptors, which are activated by one or more of the NTs. NTs preferentially bind to their respective Trk receptors. Brain-derived neurotrophic factor (BDNF), NT-4, nerve growth factor (NGF), and neurotrhphin-3 (NT-3) preferentially interact with specific Trk receptors. NGF binds to TrkA and NT-3 binds to TrkC. In the presence of the p75 neurotrophin receptor (p75^NTR^), BDNF has a high affinity for the primary ligand TrkB and interacts with it through the immunoglobulin constant 2 (Ig-C2) domain [1,2]. 

BDNF is widely distributed in the cortical areas, hippocampus, visual cortex, and in various parts of the adult CNS such as the striatum, substantia nigra (SN), retrorubral region, and ventral tegmental area (VTA), which contains a major portion of the dopaminergic (DAergic) cell groups of the ventral midbrain [3,4,5]. TrkB is highly expressed in the central nervous system, comprising the neurons of the SN pars compacta (SNpc), dorsal raphe nucleus, and VTA. TrkB is expressed in the frontal cortex, hippocampus, cerebellar cortex, pituitary gland, visual system, and hypothalamus [6,7,8,9,10,11]. The majority of the DAergic neurons of the SNpc in humans display immune reactions against TrkB (71%) and BDNF (74%). In this review, we describe the roles of BDNF/TrkB signaling, as well as those of TrkB isoforms in Parkinson’s disease (PD). 

## 2. The General Function of BDNF/TrkB Signaling in Neuron

Several lines of evidence reveal that the pleiotropic activities of BDNF and TrkB play a vital role in the survival and maintenance of DAergic neurons. BDNF protects the catecholamine biosynthetic enzyme tyrosine hydroxylase (TH)-positive nigral DAergic neurons, from the neurotoxicity of DAergic neurotoxins [12]. Following the association of BDNF with the Ig-C2 domain of TrkB, autophosphorylation of the tyrosine residues in the cytoplasmic domain of TrkB takes place, which serves as the docking site for partner proteins. This regulates the maintenance of long-term potentiation (LTP) in hippocampal CA1, and the differentiation and survival of neurons through the activation of the major phospholipase Cγ1 (PLC-γ1), Ras-mitogen-activated protein kinase (MAPK), and phosphoinositide 3-kinases (PI3K)-AKT signaling pathways [13] (Figure 1).

TrkB enhances the synaptic plasticity during both the early- and late-phase LTP in the hippocampus neurons. Following its release during LTP, BDNF stimulates the synthesis of new proteins for different temporal phases of synaptic enhancement [14]. BDNF treatment of knockout mice promoted the recovery of LTP impairment in the hippocampus [5]. Moreover, mutual and bidirectional linking between BDNF/TrkB and glutamatergic systems plays a critical role in neuroplasticity. BDNF increases glutamate release through the activation of PLCγ-mediated Ca^2+^ release [15] and regulates the signal transmission via synapses by interacting with glutamate receptors [16]. Moreover, BDNF directly or indirectly regulates glutamate signaling by regulating the expression of glutamate receptor subunit and Ca^2+^-regulating proteins or by inducing B-cell lymphoma 2 (Bcl-2) family proteins, antioxidant enzymes, and energy-regulating proteins. Conversely, glutamatergic systems lead to the stimulation of BDNF production [16,17]. Furthermore, TrkB promotes neuronal survival through the stimulation of angiogenesis. TrkB aids in the repair of the neurovasculature by enhancing endothelial survival through activation of the PI3K-AKT signaling pathway [18]. Additionally, TrkB and its ligand play a key role in learning and memory. The expression of BDNF and TrkB is associated with memory acquisition. Treatment with antisense BDNF oligonucleotide impaired memory retention as well as working memory acquisition through activation of the Ras-MAPK and PI3K-AKT signaling pathways [19,20,21]. Genetic disruption of TrkB leads to impairment in learning and memory acquisition [22]. However, recent findings suggest that the dysfunction of TrkB is also associated with neurological and psychiatric disorders.

## 3. Correlation between BDNF/TrkB Signaling and PD

PD is a neurodegenerative disorder that impairs motor or nonmotor functions. The cardinal motor dysfunction of PD is caused by the progressive degeneration of DAergic neurons in the SNpc [23]. PD leads to an almost 80% reduction in the DAergic neurons of SNpc [24]. 

Previous reports have demonstrated the high expression of BDNF and TrkB in DAergic neurons of the SN, and the enhanced maintenance, differentiation, and survival of DAergic neurons. Numerous studies have demonstrated the involvement of BDNF/TrkB signaling in PD and assessed the potential therapeutic application of BDNF. The expression of BDNF in SN was significantly lower in PD patients compared with that in control [25,26]. In Wistar male rat models of PD, extensive destruction of DAergic neurons led to a decrease in the expression of dopamine (DA) D3 receptor. However, BDNF infusion recovered the expression of DA D3 receptor in the striatum. Furthermore, BDNF knockout mice showed reduced DA D3 receptor expression [27]. BDNF protects hippocampal neurons from oxidative damage due to injury and inflammation via heme oxygenase (HO-1) induction, which is achieved by inducing Nrf2 nuclear translocation via activation of the Ras-MAPK and PI3K-AKT signaling pathways [28,29] (Figure 2). These studies suggest that the reduction in BDNF expression is linked to pathological alterations of the DAergic neurons in the SNpc.

Endoplasmic reticulum (ER) stress induces activation of the unfolded protein response (UPR) by the accumulation of misfolded proteins and eventually leads to PD by inducing the apoptosis of DAergic neuronal cells and rat cerebellar granule neurons (CGNs). Induction of UPR by ER stress is triggered by the activation of ER stress kinases, including inositol-requiring enzyme 1α (IRE1α), eukaryotic Initiation Factor 2α (eIF2α), and Protein kinase RNA-like endoplasmic reticulum kinase (PERK), and ER stress-associated proteins, 78-kDa glucose-regulated protein (GRP78) and growth arrest- and DNA damage-inducible gene 153 (GADD153), and cleavage of ER-specific procaspase-12 [30]. Additionally, ER stress induces neuronal apoptosis through the activation of Glycogen synthase kinase 3β (GSK3β), suppression of cyclin D1, and inactivation of AKT. However, TrkB overexpression activates the PI3K/AKT pathway, which in turn induces cyclin D1 expression and then prevents neuronal apoptosis due to ER stress by enhancing the phosphorylation of GSK3β at the inhibitory site (Ser9) [31]. Moreover, BDNF upregulation promotes neuronal growth through the Wnt/β-catenin signaling pathway, which induces the BDNF/TrkB pathway and reduces the expression of GSK3β [32]. Furthermore, TrkB^+/−^·5XFAD (B6SJL-Tg[AβPP *K670N*M671L*I716V*V717I, PSEN1*M146*L286V]6799Vas/J) mice showed exacerbated memory decline, significantly reduced phosphorylation of the inhibitory site (Ser9), and reduced hippocampal expression of the α-Amino-3-hydroxy-5-methyl-4-isoxazolepropionic Acid (AMPA)/N-Methyl D-aspartate (NMDA) glutamate receptor subunits [33] (Figure 3). 

The 1-Methyl-4-phenyl-1,2,3,6-tetrahydropyridine (MPTP) leads to selective damage of the neurons in the SN, which are involved in PD, and its study provides significant insight into the understanding of PD [34,35]. TrkB is widely distributed in the cytoplasm and cell membrane of DAergic neurons of the SN. In MPTP-induced C57/BL mouse models of PD, TrkB expression was significantly reduced to 36.2%, which is ~65.7% of the normal level, and the TrkB-positive DAergic neurons were more or less sensitive, owing to MPTP treatment [36]. Further, intrastriatal implantation of BDNF-secreting fibroblasts into MPTP-induced Sprague–Dawley rats increased the DA content in the SN [37].

Aging is the primary risk factor for PD. Aging leads to a reduction in DA content in the striatum and SN of humans [38], and reduction in the expression of age-related genes, including TrkB, which is involved in SN DAergic neuronal function and survival, and is associated with motor impairments in PD, which occur due to aging. The Fischer 344 (F344) rat model showed a significant decrease in TrkB expression in the DAergic neurons of SN [39]. Moreover, telomere shortening was also found to be linked to PD. In the telomere knockout (Terc^-/-^) mouse model, which plays a central role in demonstrating cell fate and aging, the expression of BDNF, TrkB, AKT, and ERK1/2 in the hippocampus and dentate gyrus were found to be downregulated [40]. 

An *α-s*ynuclein transgenic mouse with short telomeres (α-syn^tg/tg^ G3Terc^-/-^) displayed accelerated incidence of PD, and a markedly lower life span [41]. Shortened telomeres in human induced pluripotent stem cells (hiPSC), owing to the pharmacological downregulation of telomerase, showed significant loss in the expression of TH, which is a characteristic feature of early PD [42]. Progerin, a key regulator of premature aging in various tissues, induced neuronal aging and the late onset of PD phenotypes [43]. These studies suggest that the loss of BDNF/TrkB activation leads to the onset of PD.

## 4. Correlation between BDNF/TrkB and α-Synuclein

The α-Synuclein (α-syn) is a major component of Lewy bodies (LBs) and Lewy neurites (LNs). The physiological functions of α-syn are still unclear. However, they are known to play a role in intercellular DA storage, synaptic membrane biogenesis, and lipid transport [44]. Mutations, particularly A30P, A53T, and E46K, in α-syn are known to be associated with PD. These mutations lead to the formation of protofibrils, which aggregate into larger inclusion bodies [44,45].

Pathogenic α-synuclein mutations are linked to a loss in BDNF and TrkB expression. Wild-type α-syn induces BDNF expression, while the mutant of α-syn (A30 and A53T) failed to induce BDNF expression [46]. TrkB expression was also markedly reduced in the α-syn mutant, A30P transgenic mice, relative to the wild-type α-syn transgenic mice [47]. The retrograde axonal transport of BDNF/TrkB signaling endosomes was found to be essential for dendritic growth and development in cortical neurons. This occurred via the internalization of ligand-receptor complexes into endosomes, subsequently leading to retrograde signaling, which induces the transcriptional activation of the nuclear targets [48]. However, axonal transport of BDNF/TrkB is markedly impaired in neurons with axonal α-syn fibrillar aggregates [49]. Furthermore, α-syn reduces TrkB expression by interacting with the kinase domain of TrkB and inducing its ubiquitination. BDNF and Fyn inhibit TrkB degradation by blocking the formation of the α-syn-TrkB complex [50] (Figure 3). TrkB loss reduces the total neuronal numbers in the SNpc of aged TrkB^+/-^ mice, compared to the aged control mice. Interestingly, α-syn aggregation significantly accumulated in the SN of TrkB^+/-^ adult and aged mice [51]. Further studies of this functional crosstalk between BDNF/TrkB and the biomarkers of PD, like α-syn, are required to develop new effective therapies for the repair of neuronal degeneration. 

## 5. The Function of TrkB Isoform in PD

TrkB is a receptor tyrosine kinase located on chromosome 9q22 [52]. TrkB receptor initially synthesized as a precursor protein eventually yields the mature TrkB (145 kDa) protein by N-glycosylation at 10 sites in the extracellular domain [1,53]. Alternative splicing of TrkB produces different TrkB isoforms. Thirty-six potential TrkB isoforms have been identified so far. However, the probable functions of these isoforms have not yet been studied [54] (Figure 4 and Table 1). The physiological functions of most TrkB isoforms remain unclear. However, recently, several studies have demonstrated that the TrkB isoforms may be implicated in neurophysiological processes. TrkB and TrkB.T1, the most critical isoforms in the CNS, are highly expressed in the brain. TrkB.T1 has the same binding affinity to BDNF as TrkB [55]. However, TrkB.T1 is recognized as a dominant-negative receptor of TrkB and inhibits the functioning of the BDNF-TrkB signaling pathway. Recently, a new feature of TrkB.T1 has been identified. Spinal cord injury (SCI) causes neuronal death, severe neuropathic pain, impaired motor function, and loss of sensation. SCI subsequently leads to PD, owing to the accumulation of α-synuclein. Reduction in α-syn accumulation post-injury induces neuronal survival [56].

In a population-based longitudinal follow-up study, SCI was found to be associated with an increased risk of PD. A population-based and longitudinal follow-up cohort study exhibited the PD-free survival rate for the SCI patients was lower than that for the healthy group. Also, the incidence of PD development from SCI patients increased 1.65 fold relative to non-SCI groups. [57]. TrkB.T1 plays a critical role in SCI. Following SCI, TrkB.T1 was found to be overexpressed throughout the subcortical white and gray matter, and in the ependymal cells and astrocytes [58,59]. Additionally, TrkB.T1 expression was increased after SCI, and elimination of TrkB.T1 resulted in functional recovery, including the recovery of motor function, and reduction of mechanical hyperesthesia. Also, TrkB.T1 loss contributes to a reduction in neuropathic pain and decrease in SCI-induced expression of cell cycle genes [60]. Loss of TrkB.T1 in astrocytes suppressed their migration and proliferation by suppressing the expression of genes associated with inflammation, proliferation, and migration pathways. Additionally, it facilitated functional recovery, which includes reduced hyperplastic responses, and improved motor coordination following SCI [61]. Moreover, TrkB.T1 was found to be involved in the maturation of the cortical astrocytes, which is a critical process in CNS development. TrkB.T1 showed highest expression levels during the morphological maturation of astrocytes, and BDNF-TrkB.T1 signaling was found to increase the morphological complexity of astrocytes. TrkB.T1 knockout in astrocytes revealed the loss of BDNF-induced complexity, resulting in immature astrocytes with reduced volume [62].

Recent studies have demonstrated that TrkB.T1 plays a role in the development and progression of PD. The development and progression of PD and acute CNS injury, including traumatic brain injury (TBI) and SCI, are strongly linked with enhanced inflammatory response and damage [67,68]. TBI and SCI are some of the risk factors for developing PD. Patients with TBI had a 44% higher risk of developing PD with nonmotor symptoms, over the period of 5 to 7 years [69,70]. TrkB.T1 was upregulated and widely distributed in the striatum and SNpc of PD patients [55]. Various studies support the involvement of TrkB.T1 in PD. Amyloid-β (Aβ) was also found to be involved in the development of PD. The expressions of Aβ and α-syn were markedly increased in a 6-hydroxydopamine (6-OHDA)-induced mouse model of PD. Furthermore, up to 50% of PD patients with dementia (PDD), who were comorbid with Alzheimer’s disease (AD), exhibited elevated accumulation of Aβ plaques like α-syn and tau-containing neurofibrillary tangles compared with that in PD patients without dementia. Thus, AD neuropathology was linked with the pathogenesis of PDD [71]. Moreover, amyloid fibrils of α-syn, which possessed a cross-β structure, aggregated in Lewy bodies in PD patients [72]. Aβ induced the upregulation of TrkB.T1 and TrkB.T2 transcripts. Aβ also induced the calpain-mediated cleavage of TrkB and subsequently formed a new truncated TrkB receptor and an intracellular fragment of TrkB. Moreover, TrkB cleavage by calpain impairs the BDNF function [73]. Furthermore, MPTP increased the level of TrkB.T1 expression [74]. Additionally, Rbfox1 (RNA-binding protein fox-1 homolog), a neuron-specific splicing factor, was one among the most consistently upregulated genes identified in the transcriptome of midbrain dopaminergic (mDA) neurons of PD patients [75]. Rbfox1 induces TrkB.T1 expression and suppresses BDNF-induced LTP [76].

Although TrkB.T1 was found to be involved in the pathogenesis of PD through the negative regulation of BDNF/TrkB signaling, the role of other TrkB isoforms is not fully understood. In order to develop new effective treatments, further studies are required to ascertain whether the presence of TrkB isoforms is critical to the pathogenesis of PD.

## 6. BDNF/TrkB Signaling and Therapy for Parkinson’s Disease

Currently, several medications with varying degrees of activity against PD are available. These can be grouped into DA precursors, monoamine oxidase (MAO)-B inhibitors, DA agonists, anticholinergics, and glutamate antagonists.

### 6.1. Levodopa

Levodopa is a DA precursor, found in both the CNS and peripheral nervous system (PNS). It frequently combines with carbidopa, a DA decarboxylase inhibitor, which prevents levodopa from getting converted to DA, and allows it to cross the blood–brain barrier and get absorbed [77]. 

It has recently been demonstrated that BDNF expression was correlated with improvement in the clinical symptoms of PD patients. Repeated injections of levodopa caused a significant increase in BDNF mRNA levels in the subthalamic nucleus (STN) [78], thereby indicating that levodopa treatment produces a dose-dependent upregulation of BDNF expression [79]. 

BDNF has been proposed as a putative candidate for the development of prolonged levodopa-induced dyskinesia (LID). BDNF overexpression in the 6-OHDA-lesioned Sprague–Dawley mouse model of PD was associated with LID, caused by interventions in serotonin neurons. BDNF overexpression induces significant striatal serotonin terminal sprouting, thereby subjecting the neurons to develop LID and levodopa-induced rotations [80,81]. However, another report suggested that BDNF expression had no correlation with dyskinesia. CI-1041 (Besonprodil) or cabergoline treatment was found to prevent chronic LID. However, no difference was seen in BDNF expression in long-term levodopa-treated monkeys, compared to those treated with a combination of levodopa and CI-1041 (N-Methyl-D-Aspartate Receptor Subunit 1A/2B N-methyl-D-aspartate (NR1A/2B NMDA) receptor antagonist) or levodopa and cabergoline (DA D2 receptor agonist). Also, long-term levodopa therapy was associated with a reduction in BDNF concentration in the anterior and posterior caudate nucleus of monkeys [82]. The involvement of BDNF expression in LID is not fully understood and requires further investigation.

### 6.2. Neupro (Rotigotine)

Neupro (rotigotine) is a non-ergolinic DA receptor agonist, delivered slowly across the skin by a transdermal patch. It is used for the treatment of early- or advanced-stage PD symptoms and restless legs syndrome (RLS). It acts as a DA substitute in the brain. Studies have shown that Neupro exhibits the highest affinity for the DA receptor [83,84,85]. Formation of D1-D2 DA receptor heteromer as a result of Neupro treatment increases intracellular Ca^2+^ and phospholipase C (PLC) levels and induces the activation of calcium/calmodulin-dependent kinase IIα (CaMKIIα). Subsequently, it induces BDNF expression in the cortex and hippocampus by controlling the activation of BDNF promoter I and IV through regulation of neuronal activity and intracellular Ca^2+^ concentration [86]. 

### 6.3. Selegiline (Deprenyl)

Selegiline (deprenyl), an irreversible MAO-B inhibitor, is usually used as a monotherapy for early-stage PD treatment. Selegiline treatment markedly recovered MPTP-mediated fore- and hindlimb stride length, and attenuated the loss of TH-positive nigral neurons and striatal axons. Furthermore, selegiline rescued motor function deficit via induction of the expression of anti-apoptotic factors and Glial cell-derived neurotrophic factor (GDNF)/BDNF in the SNpc of MPTP-exposed mice [87]. A meta-analysis of a combination of selegiline and levodopa showed an improvement in the clinical symptoms of PD patients, owing to enhanced drug effect, compared to monotherapy of selegiline. Combination therapy was demonstrated to prolong the effectiveness of levodopa, reduce the amount of levodopa, reduce fluctuations in motor or nonmotor functions, and improve the effectiveness and quality of life [88,89]. Moreover, selegiline treatment was shown to induce the expression of oxidative stress-related proteins such as HO-1, PrxI, TrxI, TrxRxI, γGCS, and p62/A170. TrkB-mediated PI3K activation by selegiline exhibited increased cytoprotective and antioxidant effects in PD patients through the induction of HO-1 expression by increasing the nuclear retention of Nrf2 [90].

### 6.4. Azilect (Rasagiline)

Azilect (rasagiline) is a MAO-B inhibitor, which increases the levels of DA in the brain. The Azilect treatment triggers neuro-protective activities such as the induction of anti-apoptotic Bcl-2 family proteins and NTs (NGF, BDNF, and glial cell line-derived (GDNF)) and activation of the PI3K-AKT survival pathway [91]. It interferes with the interaction between α-syn and TrkB and, subsequently, prevents the loss of DAergic neurons by restoring the function of the BDNF/TrkB signaling pathway [50]. Furthermore, rasagiline treatment suppresses α-syn/TrkB complex formation by inhibiting the production of the MAO-B-mediated DA metabolite, 3,4-dihydroxyphenylacetaldehyde (DOPAL) in the SNpc of α-Syn- or MPTP-induced PD mouse models. This suppression leads to the subsequent activation of TrkB downstream signaling pathways such as Ras-MAPK and PI3K-AKT [50].

### 6.5. Memantine

Memantine is a noncompetitive NMDA receptor antagonist against glutamate-mediated neurotoxicity. It is used in the treatment of moderate to severe AD and functions by improving memory, awareness, and the ability to perform daily activities. Memantine also has neuroprotective functions in PD patients. Patients treated with memantine show moderate-to-substantial improvement of symptoms [92,93]. Moreover, memantine treatment restores the MPTP-mediated reduction of BDNF expression and phospho-TrkB levels [94].

### 6.6. Amantadine

Amantadine is a glutamate and NMDA receptor antagonist used in the treatment of the first stage of PD or LID. It is used as monotherapy, or as a combination therapy with levodopa or DA agonist. Amantadine treatment reduces the toxicity of DAergic neurons by 1-methyl-4-phenylpyridinium ion (MPP+) or lipopolysaccharide (LPS) via the inhibition of microglial pro-inflammatory factor release and induction of GDNF expression [95]. A meta-analysis of amantadine treatment exhibited reduced LID [96]. Induction of BDNF expression by amantadine in the hippocampus contributed to its antidepressant activity [97]. Moreover, cotreatment of fluoxetine and amantadine significantly induced BDNF expression in the cerebral cortex [98].

### 6.7. Cell Replacement Therapy

Although symptoms of PD patients can be alleviated by surgical or pharmaceutical treatments in the early stages, these therapies have serious side effects. They can lead to PD-related complications like depression and anxiety disorders, dopamine dysregulation syndrome, impulse control disorders, psychosis, and manic syndromes [99]. For these reasons, the therapeutic potential of stem cell therapy has been explored for the replacement of damaged neurons in PD patients. Stem cells are well characterized by their ability to self-renew and differentiate into any cell type.

Neural transplantation therapy using embryonic stem cells has been applied in clinical trials of human neurodegenerative diseases like AD, PD, Huntington’s disease, epilepsy, and strokes [100]. Transplantation of testicular Sertoli cells into the brain enhances the regeneration and promotes the survival of the grafted DAergic neurons. DAergic neuron differentiation from Sertoli-induced primate embryonic stem (ES) cells induces markers of mature DAergic neuronal phenotypes (TH, DA transporter (DAT), aromatic amino acid decarboxylase (AADC)), and transcription factors such as Nurr1 and Lmx1b, and Trk receptors (TrkA, TrkB, and TrkC) [101].

Transplantation of the induced pluripotent stem cells (iPSCs) of the mouse into PD rats results in the differentiation of the iPSCs into Glia cells and neurons, which in turn gets functionally integrated into the host brain, and improves the behavior of the mouse model of PD [102].

Human iPSCs from the skin of idiopathic PD patients can differentiate into DAergic neurons [103]. Transplantation of human iPSCs into a 6-OHDA-induced Fischer 344 mouse model of PD resulted in the differentiation of the iPSCs into DAergic neurons in the midbrain (mDA) and expression of TrkB on the surface of mDA-specific human iPSCs [104]. The iPSCs from a healthy donor, on transplantation into PD patients carrying α-syn with A53T mutation, were well differentiated to human neuronal precursor cells (hNPCs) that efficiently internalized α-syn fibrils, which were then degraded efficiently via the lysosomal pathway within three days after internalization. The hNPCs also efficiently transfer α-syn fibrils to the lysosomal vesicles between them, through the formation of numerous tunneling nanotube (TNT)-like structures between the hNPCs [105,106]. However, the use of stem cells can have adverse effects as well as ethical issues. Human umbilical cord-derived mesenchymal stem cells (UC-MSCs) are widely used as an ethically acceptable source of SC, can be obtained without any risk to the donor, and are rarely contaminated by infectious agents [107]. UC-MSCs derived from the mesoderm possess strong proliferation ability, and multiple differentiation potential. Differentiation of UC-MSCs increased the number of immature or mature neuron-like cells and DA neuron-like cells. It induced BDNF and TrkB expression and increased DA release or the number of DAT- and TH-positive cells. Also, transplantation of UC-MSCs into rats showed the possibility of prolonged survival, as well as significant behavioral recovery [108,109]. These studies supported the possibility of conducting a clinical trial, and in January 2018 the use of intravenous infusion of UC-MSC therapy to treat PD was under Phase I clinical trial (NCT03550183).

### 6.8. Other Agents

Omega-3 polyunsaturated fatty acids (n-3 PUFAs) have a neuroprotective effect in Tg2576 mouse models of AD. Low n-3 PUFA consumption was found to be associated with a higher risk of developing AD [110,111]. Dietary intake of n-3 PUFA improves neuroprotective activities in the MPTP-induced C57BL/6 mouse model of PD. Treatment with n-3 PUFAs markedly increased BDNF and TrkB expression in the motor cortex of the MPTP-induced C57BL/6 mouse model of PD [112]. 

The 7,8-Dihydroxyflavone (7,8-DHF), a TrkB agonist, has been reported to activate TrkB and its downstream signaling cascades in order to promote cell survival/growth, differentiation, and plasticity. Treatment with 7,8-DHF restored motor function deficits through the induction of TrkB and activation of extracellular signal-regulated kinase 1/2 (ERK1/2) in both the striatum and SN of the MPTP-induced C57BL/6 mouse model of PD [113]. Additionally, 7,8-DHF treatment aids in neuroprotection via other mechanisms. The 7,8-DHF treatment suppresses glutamate-mediated glutathione depletion. Moreover, 7,8-DHF therapy, in the absence of TrkB, protects neurons from reactive oxygen species (ROS)-induced cell death by glutamate [114] (Figure 2). Moreover, 7,8-DHF treatment in a MPTP-mediated mouse model of PD blocked the striatal terminal loss by sustaining almost 54% of TH expression in the dorsolateral (DL) striatum and increasing the level of ganglion-10 (SCG10), TrkB phosphorylation, and ERK1/2 phosphorylation within the striatum and SN of MPTP-mediated C57BL/6J mouse model of PD [115]. The activation of TrkB by 7,8-DHF has been shown to mediate TrkB glycosylation. The N-glycosylated extracellular domain of TrkB strongly interacts with 7,8-DHF and, subsequently, triggers the internalization of the TrkB receptor through clathrin-mediated endocytosis [116]. Furthermore, 7,8-DHF treatment suppresses MPTP-induced oxidative stress by the upregulation of glutathione (GSH) and superoxide dismutase (SOD) activities and inhibits MPTP-induced *α*-synuclein expression [117]. The effect of 7,8-DHF treatment was assessed in nonrodent models. In monkeys, 7,8-DHF metabolized into its major metabolite, 7-hydroxy-8-methoxy flavone, and drastically activated TrkB. Subsequently, it portrayed a neuro-protective effect on the DAergic neurons in monkeys with intracerebroventricular injections of MPP^+^. MPP^+^ treatment led to the loss of 40–95% of DAergic neurons, relative to the treatment of 7, 8-DHF, or wild-type monkeys [118]. These results indicate that 7,8-DHF treatment suppresses the accumulation of *α*-synuclein, and reduces oxidative stress through the induction of TrkB activation and subsequently blocks the loss of DAergic neurons in the SN and striatum of PD patients. Although 7,8-DHF has a neuroprotective function with no detectable toxicity in PD patients [119], it has only modest oral bioavailability and a moderate pharmacokinetic profile. To overcome the poor oral bioavailability, the pro-drug R13, a derivative of 7,8-DHF, was developed by the modification of 7,8-DHF. Pro-drug R13 exhibited good absorption by maintaining the release of 7,8-DHF into the circulation system. It has a very long half-life and improved oral bioavailability. These effects of pro-drug R13 can block synaptic loss though the sustainable induction of TrkB and its downstream signals [120].

The effect of 7,8-DHF was assessed on a rotenone-induced Lewis rat model of PD. Rotenone treatment induced PD in the animal model by triggering multiple pathogenic pathways like oxidative stress, aggregation of alpha-synuclein, and Lewy pathology [121]. However, the treatment of 7,8-DHF improved behavioral performance by the activation of TrkB and inhibited tauopathy and α-synucleinopathy [122]. Additionally, a dehydroepiandrosterone (DHEA) analog has been developed as a potential therapeutic agent for PD patients. 17-beta-spiro-[5-androsten-17,2’-oxiran]-3beta-ol (BNN-20) is a synthetic analog of the endogenous neurosteroid DHEA. It induces the phosphorylation of TrkA, TrkB, and p75^NTR^ and BDNF expression in the genetic model (Weaver) of PD, which exhibits progressive DAergic neurodegeneration in the SN [123].

## 7. BDNF and TrkB Mutation and PD

Mutations in the BDNF gene have been found to play a critical role in the development of LID. PD patients with BDNF V66M or M66M alleles demonstrated a significantly higher risk of developing LID earlier compared with PD patients with the BDNF V66V genotype. BDNF V66M or M66M mutation significantly correlated with the survival of PD patients. PD patients with BDNF V66M or M66M allele display a more reduced overall survival, compared to those with the BDNF V66V allele [124]. Based on haplotype variants in a large community-based study of PD patients, BDNF V66M (single nucleotide polymorphism (SNP) rs6265) was found to be associated with increased susceptibility to LID [125]. 

## 8. Conclusions

As described in this review, increasing evidence indicates that BDNF and TrkB have tremendous therapeutic potential for the treatment of PD. Drug-induced or cell replacement therapy-induced recovery of BDNF and TrkB expression exerts potent effects on the progressive degeneration of DAergic neurons and BDNF. TrkB expression is an intriguing candidate in the development and progression of PD. TrkB.T1, a potential therapeutic target, may play a critical role in the development and progression of PD. However, acquired results still remain mostly unknown, since the upregulated TrkB.T1 is widely distributed in the striatum and SNpc of PD patients [55]. Given the importance of the involvement of TrkB.T1 in PD, further work is needed to elucidate the function of TrkB.T1 and to develop a drug for the regulation of TrkB.T1 expression. The upregulation of TrkB.T1 regulates locomotor dysfunction and neuropathic pain in SCI and is widely distributed in the brain of PD patients. Ongoing research aims to address these questions and raises several other questions. How does TrkB.T1 involve in the development and progression of PD? Is TrkB.T1 expression correlated with α-syn, which is mainly associated with the pathology of neurodegenerative diseases like AD, PD, and Huntington’s disease? These questions need to be addressed by further studies on the role of TrkB.T1 in regulating the mechanism of α-syn-induced pathogenic pathways of neurodegenerative diseases. 

Further understanding of the mechanism of BDNF-TrkB or BDNF-TrkB.T1 signaling, and their regulatory functions in the survival and maintenance of DAergic neurons, will provide a novel and practical approach to study the pathology of neurodegenerative diseases like AD, PD, and Huntington’s disease, and investigate potential therapeutic strategies. 

## Figures and Tables

**Figure 1 jcm-09-00257-f001:**
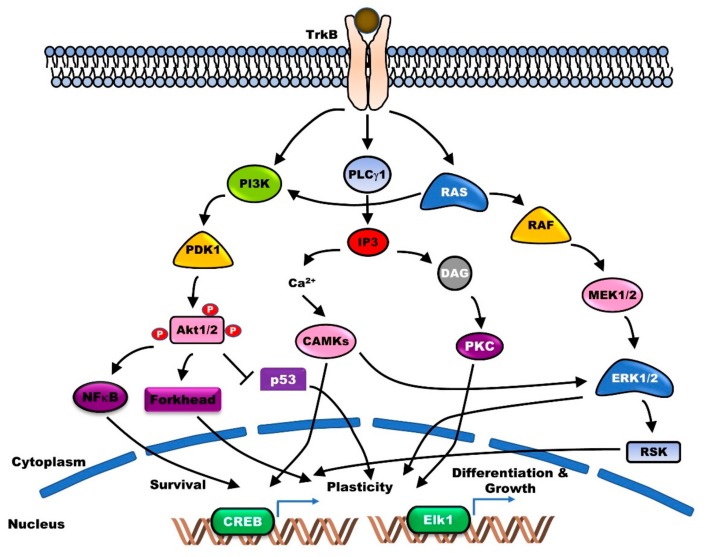
Brain-derived neurotrophic factor (BDNF)/tropomyosin-related kinase receptor type B (BDNF/TrkB) signaling supports neuronal survival, plasticity, differentiation, and growth via activation of several functional downstream cascades. Binding BDNF to TrkB as its specific receptor leads to homodimerization and triggers activation of adaptor proteins such as polypyrimidine tract-binding protein (PTB) and Src homology domain 2 (SH2). Subsequently, activated adaptor proteins lead to activation of phosphoinositide 3-kinases (PI3K)-AKT (PI3K-AKT), Ras-mitogen-activated protein kinase (Ras-MAPK), and phospholipase Cγ1 (PLC-γ1)-protein kinase C (PKC) signaling pathway.

**Figure 2 jcm-09-00257-f002:**
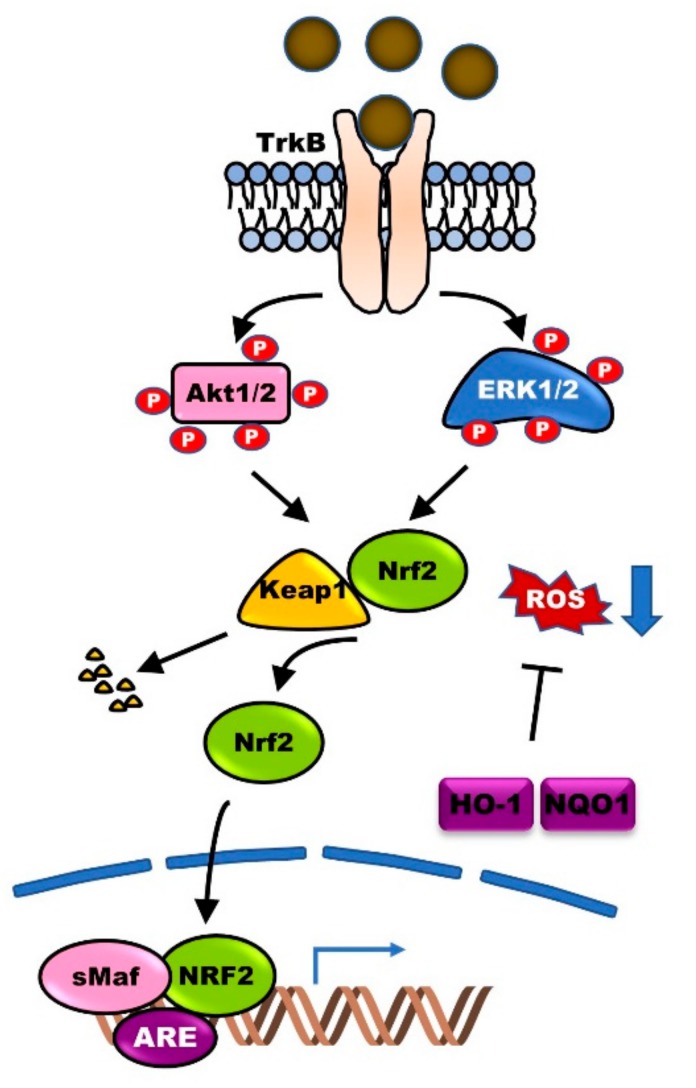
BDNF-TrkB signaling protects neurons from reactive oxygen species (ROS)-induced cell death. BDNF-TrkB signaling leads to the activation of phosphoinositide 3-kinases (PI3K)-AKT (PI3K-AKT) and Ras-mitogen-activated protein kinase (Ras-MAPK) pathways. Activated PI3K-AKT and Ras-MAPK pathways trigger dissociation of nuclear factor erythroid 2-related factor 2 (Nrf2)-Keap1 complex and then induces nuclear translocation of Nrf2. Finally, the binding of Nrf2 to antioxidant response element (ARE) in target genes leads to the expression of antioxidant enzymes, including heme oxygenase-I (HO-1) and, subsequently, involved in protection from ROS-mediated neuronal cell death of Parkinson’s Disease (PD).

**Figure 3 jcm-09-00257-f003:**
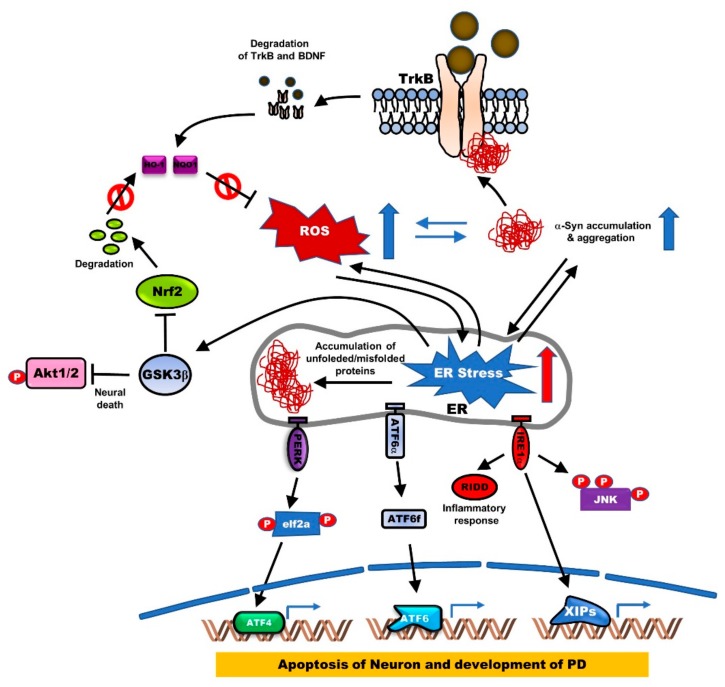
The generation of endoplasmic reticulum (ER) stress by ROS and accumulation of α-syn aggregates disrupts BDNF-TrkB-mediated neuronal protection. Accumulation or mutation of α-syn in PD reduces BDNF or TrkB expression through TrkB ubiquitination, and its reduction of BDNF-TrkB signaling leads to the generation of ER stress by induction of ROS or accumulation of α-syn. ER stress triggers the misfolding of proteins, leading to the accumulation of protein aggregates. Accumulation of protein aggregates activates protein kinase RNA-like ER kinase (PERK), activating transcription factor (ATF6α), and inositol-requiring protein (IRE1α) as three main transmembrane proteins in the ER. The activation of PERK, which induces eukaryotic Initiation Factor 2α (eIF2α) phosphorylation, inhibits protein translation and increases ATF4, which is involved in apoptosis through induction of the CCAAT-enhancer-binding protein (C/EBP) homologous protein (CHOP) known as growth arrest and DNA damage-inducible protein. Also, activated IRE1α induced neuronal apoptosis by induction of c-Jun N-terminal kinase (JNK) phosphorylation, or by increasing inflammatory response inducting IRE1α-dependent messenger RNA decay (RIDD). Additionally, the production of ATF6 fragment (ATF6f) by cleavage of ATF6 in the Golgi apparatus is involved in the development of PD by inducing apoptosis of neuronal cells.

**Figure 4 jcm-09-00257-f004:**
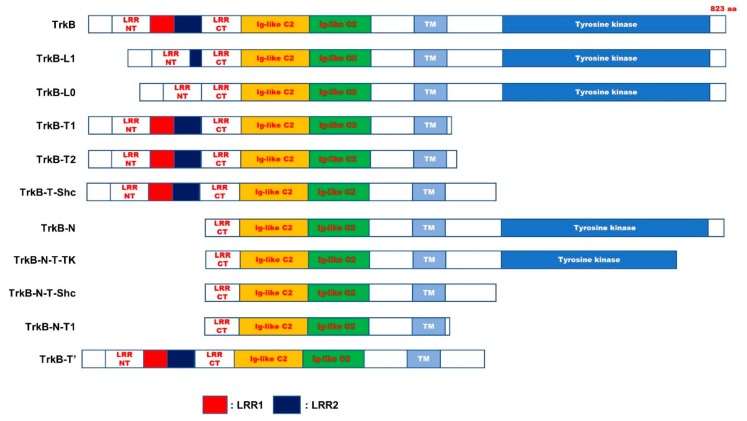
Schematic representation of TrkB full length and truncated isoforms. The domain of TrkB exhibited as filled boxes. TrkB-N, TrkB-N-T-TK, TrkB-N-T-Shc, and TrkB-N-T1 have a lack of N-terminal signal sequence, leucine-rich repeat N-terminal domain (LRRNT), leucine-rich repeat region 1 (LRR1), and LRR2 domain of TrkB. Also, TrkB-L1 and TrkB-L0 lacked the first two or three or all three of LRRs in the extracellular domain of TrkB, respectively. Additionally, TrkB.T1 and TrkB.T2, TrkB-T-Shc, TrkB-T’ lack c-terminal region of TrkB, including tyrosine kinase domain. LRRCT, leucine-rich repeat C-terminal domain; Ig-like C2, immunoglobulin-C2-set domain; TM, transmembrane domain.

**Table 1 jcm-09-00257-t001:** Characterization of TrkB isoforms.

Name	Characterization	Relevance of PD	
TrkB-L1	lacked the first two LRMs of three leucine-rich motifs (LRMs) in the ECD	ND	[63]
TrkB-L0	lacked the all of three leucine-rich motifs (LRMs) in the ECD	ND	[63]
TrkB.T1	467-477 AA: PASVISNDDDS → FVLFHKIPLDG 478-822 AA: Missing	Involved	[64]
TrkB.T2	Contain only 23 amino acids of the ICD.	Involved	[64]
TrkB-T-Shc	529-537 AA: FVQHIKRHN → WPRGSPKTA 538-822 AA: Missing.	ND	[54,65]
TrkB-T-TK	710-735 AA: GGHTMLPIRWMPPESIMYRKFTTESD → SSCADQRPQGPLSLRDPCCICLLRLS 736-822 AA: Missing.	ND	[54,65]
TrkB-N	lack of N-terminal signal sequence	ND	[65]
TrkB-N-T-TK	710-735 AA: GGHTMLPIRWMPPESIMYRKFTTESD → SSCADQRPQGPLSLRDPCCICLLRLS 736-822 AA: Missing.	ND	[54,65]
TrkB-N-Shc	529-537 AA: FVQHIKRHN → WPRGSPKTA 538-822 AA: Missing.	ND	[54,65]
TrkB-N-T1	1-156 AA: Missing. 467-477 AA: PASVISNDDDS → FVLFHKIPLDG 478-822 AA: Missing.	ND	[54,65]
TrkB.Kin	Additional six-AA insertion between the Ig2 and the TM	ND	[66]

TM, transmembrane region; ICD, intracellular domain; ECD, extracellular domain; TKD, tyrosine kinase domain; LRRNT, leucine-rich repeat N-terminal domain; LRR1, leucine-rich repeat region 1; LRR2, leucine-rich repeat region 1; ND, not determined.

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
