# Peer review of "Regulation of BDNF-TrkB Signaling and Potential Therapeutic Strategies for Parkinson’s Disease"

_jcm, 2020, doi:10.3390/jcm9010257_

Round 1

Reviewer 1 Report

It is clear the purpose of this review is to provide a comprehensive overview of the role of BDNF/TrkB signalling in Parkinson's Disease (PD), with an extensive amount of literature incorporated and described in this manuscript.The effort to put this review together should be commended.  For most of the review, particularly in early sections, I was not able to identify a clear, direct or unified perspective on whether BDNF/TrkB signalling is helpful, harmful, or irrelevant in PD or whether it should/should not be targeted in terms of therapeutics for PD. Instead, this manuscript is largely descriptive. In a review article, I would expect a critical appraisal of the literature, identification of the key knowledge gaps and potential ways to overcomes these to be identified and discussed with a cogent and cohesive argument to be easily identified throughout.

Specific comments to be addressed in order to improve the manuscript are below:

Section 3 -- there needs to be greater clarity when referring to animals models of PD. This includes strengths/weaknesses of specific PD animal models, and it needs to be easily distinguished when data being discussed is generated from an animal model or the human disease. Relevant to Lines 72-80. In general, reviews are heavily accessed by non-experts and should be understood by them.

Lines 90-93 are isolated. What is the relevance of this to either PD or the mechanistic roles of BDNF/TrkB in PD?

Lines 94-110; again the links between ER stress, the 6-OHDA model and TrkB signalling are not clearly articulated and there is no cohesive argument. There is no comment on how opposing signaling outcomes can interact/co-exist and the relevance of this is PD pathophysiology is not clearly articulated. 

Line 103-104; why are these methodological aspects are important to understand the context of the literature?

Line 105-106; MPTP model is introduced with no context. How does it relate to the human disease or other PD animal models? Why is what happens to TrkB expression in this model important?

Line 110; similarly, what is the telomere deficient animal model? How does it relate PD? Why is the change in BDNF or other cellular signalling pathways (which may not be solely BDNF-TrkB mediated) important to understand?

This section is missing a summary statement that answers the question: What is the collective evidence pointing to in terms of the role BDNF/TrkB has in PD? 

Section 4 --

Lines 140-145; Explanation is poor and difficult to understand

Line 147; what are 'these interventions' that require further study? It is not clear what the author means here.

Section 5 --

Description of the different TrkB isoforms should be summarised succinctly and the relevance of specific isoforms to PD clearly identified. The current form reads as a shopping list and is difficult to identify what the important information is.

Relevance of spinal cord injury (SCI) context to PD is not articulated. Anatomical location of isoform expression is likely to be relevant to function and this is not discussed, or noted. Greater precision in discussion of why these different TrkB isoforms are relevant to PD is needed. 

Line 201 "TrkB.B1 may be involved in the development of PD." How? This is not drawn out and there is no evidence for this discussed/presented in the text, all evidence is drawn from TBI/SCI which are different neurodegenerative contexts. This argument is not well made. 

Section 6 & 7 -- many subsections here, do not contain a cohesive argument or perspective on the relevance of BDNF/TrkB signalling in either these treatment agents. Largely, because there has been no discussion of the link between BDNF and glutamergic signalling. I would recommend that this be discussed in Section 2 to provide background to understand some of these arguments.

Author Response

Reviewer #1 Comments and Suggestions for Authors

It is clear the purpose of this review is to provide a comprehensive overview of the role of BDNF/TrkB signalling in Parkinson's Disease (PD), with an extensive amount of literature incorporated and described in this manuscript.The effort to put this review together should be commended.  For most of the review, particularly in early sections, I was not able to identify a clear, direct or unified perspective on whether BDNF/TrkB signalling is helpful, harmful, or irrelevant in PD or whether it should/should not be targeted in terms of therapeutics for PD. Instead, this manuscript is largely descriptive. In a review article, I would expect a critical appraisal of the literature, identification of the key knowledge gaps and potential ways to overcomes these to be identified and discussed with a cogent and cohesive argument to be easily identified throughout.

We thank the Reviewer for this comment. As suggested by the Reviewer, we have corrected and modified the sentence in the Abstract to make it more transparent regarding the role of BDNF/TrkB signaling in PD. Also, the manuscript has been edited by an English editing Service.

Specific comments to be addressed in order to improve the manuscript are below:

Section 3 -- there needs to be greater clarity when referring to animals models of PD. This includes strengths/weaknesses of specific PD animal models, and it needs to be easily distinguished when data being discussed is generated from an animal model or the human disease. Relevant to Lines 72-80. In general, reviews are heavily accessed by non-experts and should be understood by them.

We thank the Reviewer for this comment. As suggested by the Reviewer, we have corrected and modified the sentence in the entire manuscript ,including Section 3, to make it more precise and greater clarity and to easily distinguished.

Lines 90-93 are isolated. What is the relevance of this to either PD or the mechanistic roles of BDNF/TrkB in PD?

We appreciate your comment. When we checked again, we only found the phenomenon that acupuncture treatment markedly recovered BDNF expression from the reduction of stress-mediated BDNF expression in the hippocampus of the mouse PD model. So, we deleted these sentences to enhance the completeness and professionalism of our manuscript.

Lines 94-110; again the links between ER stress, the 6-OHDA model and TrkB signalling are not clearly articulated and there is no cohesive argument. There is no comment on how opposing signaling outcomes can interact/co-exist and the relevance of this is PD pathophysiology is not clearly articulated. 

We really appreciate your comment. As suggested by the Reviewer, we have corrected and modified the sentences in Lines 101-114 to clarify the relevance between ER stress and BDNF/TrkB signaling in PD.

Line 103-104; why are these methodological aspects are important to understand the context of the literature?

We thank the Reviewer for this comment and agree with the Referee’s opinion and suggestion. We removed methodological aspects.

Line 105-106; MPTP model is introduced with no context. How does it relate to the human disease or other PD animal models? Why is what happens to TrkB expression in this model important?

We thank the Reviewer for this comment. As suggested by the Reviewer, we have added, corrected and modified the sentences in Line 115-121 to explain how MPTP treatment used as the PD model and what happens to TrkB expression in this model.

Line 110; similarly, what is the telomere deficient animal model? How does it relate PD? Why is the change in BDNF or other cellular signalling pathways (which may not be solely BDNF-TrkB mediated) important to understand?

We thank the Reviewer for this comment. As suggested by the Reviewer, we have added, corrected and modified the sentences in Line 122-151 to explain how telomere relates to PD. Although other cellular signaling pathways involved in the PD, here we focus on BDNF/TrkB signaling as the most frequently implicated signaling pathway in PD.

This section is missing a summary statement that answers the question: What is the collective evidence pointing to in terms of the role BDNF/TrkB has in PD? 

We added the summary statement with the sentence suggested by the reviewer.

Section 4 --

Lines 140-145; Explanation is poor and difficult to understand

We thank the Reviewer for this comment, and we apologized for the inconvenience. We corrected and modified the sentence to understand easily (Lines 159-174).

Line 147; what are 'these interventions' that require further study? It is not clear what the author means here.

We apologize for the error. We have corrected and modified the sentence.

Section 5 --

Description of the different TrkB isoforms should be summarised succinctly and the relevance of specific isoforms to PD clearly identified. The current form reads as a shopping list and is difficult to identify what the important information is.

As suggested by the Reviewer, we made a Table 1 and organized it to summarize succinctly. Also, we have added, corrected and modified the sentences in Section 5 to more clarify the relevance between TrkB isoform and pathology of PD.

Relevance of spinal cord injury (SCI) context to PD is not articulated. Anatomical location of isoform expression is likely to be relevant to function and this is not discussed, or noted. Greater precision in discussion of why these different TrkB isoforms are relevant to PD is needed. 

We thank the Reviewer for this comment. Several studies demonstrated that spinal cord injury is associated with subsequent increased risk of PD. SCI is associated with a subsequent increased risk of PD by the present population-based longitudinal follow-up study. A population-based and longitudinal follow-up cohort study exhibits the PD-free survival rate for the SCI patients was lower than that for the healthy group. Also, the incidence of PD development from SCI patients increases 1.65 fold relative to non-SCI groups [1]. Moreover, SCI leads to the accumulation of a-syn, which is a biomarker for PD and AD. Reduction of post-injury synuclein accumulation induced neuronal survival [2]. Furthermore, spinal cord stimulation improves walking, decreased movement symptoms, and freezing of advanced PD patients [3].

Also, according to your suggestion, we have added, corrected and modified the sentences in Section 5 to explain the function of TrkB isoform in PD.

Yeh, T.S.; Huang, Y.P.; Wang, H.I.; Pan, S.L. Spinal cord injury and Parkinson's disease: a population-based, propensity score-matched, longitudinal follow-up study. Spinal Cord 2016, 54, 1215-1219, doi:10.1038/sc.2016.74. Fogerson, S.M.; van Brummen, A.J.; Busch, D.J.; Allen, S.R.; Roychaudhuri, R.; Banks, S.M.; Klarner, F.G.; Schrader, T.; Bitan, G.; Morgan, J.R. Reducing synuclein accumulation improves neuronal survival after spinal cord injury. Exp Neurol 2016, 278, 105-115, doi:10.1016/j.expneurol.2016.02.004. Samotus, O.; Parrent, A.; Jog, M. Spinal Cord Stimulation Therapy for Gait Dysfunction in Advanced Parkinson's Disease Patients. Movement Disord 2018, 33, 783-792, doi:10.1002/mds.27299.

Line 201 "TrkB.B1 may be involved in the development of PD." How? This is not drawn out and there is no evidence for this discussed/presented in the text, all evidence is drawn from TBI/SCI which are different neurodegenerative contexts. This argument is not well made. 

We apologize for the error. TrkB.B1 is TrkB.T1. So, we have corrected and modified the sentence.

Section 6 & 7 -- many subsections here, do not contain a cohesive argument or perspective on the relevance of BDNF/TrkB signalling in either these treatment agents. Largely, because there has been no discussion of the link between BDNF and glutamergic signalling. I would recommend that this be discussed in Section 2 to provide background to understand some of these arguments.

We thank the Reviewer for this comment. As your suggestion, we added contents regarding relevance between the glutamatergic system and BDNF in Line 61-68 of Section 2.

Reviewer 2 Report

In the current review by Dr. Jin, the author has done decent job in summarizing the current concept regarding BDNF/TrkB signaling and its potential biological functions in neurons and its role in Parkinson's disease. Overall, the review is well organized. However, the section 5 on TrkB isoform in PD may need more work to clarify each of the confirmed TrkB isoforms' potential pathological roles in neurodegenerative diseases including PD. The predictive but not confirmative isoforms may be covered by proposing some presumption. Notably, the manuscript needs to be proof read by a native English speaker before publication.

Author Response

Reviewer #2 Comments and Suggestions for Authors

In the current review by Dr. Jin, the author has done decent job in summarizing the current concept regarding BDNF/TrkB signaling and its potential biological functions in neurons and its role in Parkinson's disease. Overall, the review is well organized. However, the section 5 on TrkB isoform in PD may need more work to clarify each of the confirmed TrkB isoforms' potential pathological roles in neurodegenerative diseases including PD. The predictive but not confirmative isoforms may be covered by proposing some presumption. Notably, the manuscript needs to be proof read by a native English speaker before publication.

Thank you for your assistance. We apologize for the grammatical errors and have edited our manuscript again. We have corrected and modified the sentence by the English language editing service, as suggested by the Reviewer. Also, we have corrected and modified the sentences in Section 5 to more clarify the relevance between TrkB isoform and pathology of PD. Moreover, we added the contents of the Reviewer’s suggestion containing some presumption of isoforms in Section 5.